# Suppression of the Reactive Oxygen Response Alleviates Experimental Autoimmune Uveitis in Mice

**DOI:** 10.3390/ijms21093261

**Published:** 2020-05-05

**Authors:** Sheng-Min Hsu, Chang-Hao Yang, Yu-Ti Teng, Hsien-Yang Tsai, Chieh-Yu Lin, Chia-Jhen Lin, Chi-Chang Shieh, Shun-Hua Chen

**Affiliations:** 1Department of Ophthalmology, National Cheng Kung University Hospital, College of Medicine, National Cheng Kung University, Tainan 704, Taiwan; shengmin@mail.ncku.edu.tw (S.-M.H.); andyteng1026@gmail.com (Y.-T.T.); towawa1206@gmail.com (C.-J.L.); 2Department of Ophthalmology, National Taiwan University Hospital, College of Medicine, National Taiwan University, Taipei 100, Taiwan; chyangoph@ntu.edu.tw; 3Department of Ophthalmology, Tzu Chi Hospital, Taichung 427, Taiwan; tc1512901@tzuchi.com.tw; 4Department of Microbiology and Immunology, College of Medicine, National Cheng Kung University, Tainan 701, Taiwan; sweet0720candy@hotmail.com; 5Institute of Clinical Medicine, College of Medicine, National Cheng Kung University, Tainan 704, Taiwan

**Keywords:** reactive oxygen species (ROS), experimental autoimmune uveitis (EAU), NF-κB, N-acetylcysteine (NAC), neutrophil cytosolic factor 1 (Ncf1)

## Abstract

Reactive oxygen species (ROS) are produced by host phagocytes and play an important role in antimicrobial actions against various pathogens. Autoimmune uveitis causes blindness and severe visual impairment in humans at all ages worldwide. However, the role of ROS in autoimmune uveitis remains unclear. We used ROS-deficient (*Ncf1^−/−^*) mice to investigate the role of ROS in experimental autoimmune uveitis (EAU). Besides, we also used the antioxidant N-acetylcysteine (NAC) treatment to evaluate the effect of suppression of ROS on EAU in mice. The EAU disease scores of *Ncf1^−/−^* mice were significantly lower than those of wild-type mice. EAU induction increased the levels of cytokines (interleukin (IL)-1α, IL-1β, IL-4, IL-6, IL-12, IL-17, and tumor necrosis factor (TNF)-α) and chemokines (monocyte chemoattractant protein (MCP)-1) in the retinas of wild-type mice but not in those of *Ncf1^−/−^* mice. EAU induction enhanced the level of NF-κB activity in wild-type mice. However, the level of NF-κB activity in *Ncf1^−/−^* mice with EAU induction was low. Treatment with the antioxidant NAC also decreased the severity of EAU in mice with reduced levels of oxidative stress, inflammatory mediators, and NF-κB activation in the retina. We successfully revealed a novel role of ROS in the pathogenesis of EAU and suggest a potential antioxidant role for the treatment of autoimmune uveitis in the future.

## 1. Introduction

Uveitis is among the most important causes of blindness and severe visual impairment worldwide. Approximately 15% to 30% of uveitis occurs in the choroid and adjacent retina and is therefore classified as posterior uveitis or uveoretinitis [1]. Posterior uveitis tends to damage photoreceptor cells and leads to permanent blindness. According to epidemiological data from the United States of America, uveitis occurs in approximately 0.54% of the population, in which approximately 30% of cases of uveitis are idiopathic [2]. An autoimmune causality is supported by strong human leukocyte antigen (HLA) associations and by frequent responses to one or more unique retinal antigens. In addition, uveitis is often associated with autoimmune or inflammatory disorders, such as Behcet’s disease, ankylosing spondylitis, sarcoidosis, psoriatic arthritis, Crohn’s disease, and ulcerative colitis in patients [2]. Ocular trauma may precipitate uveitis, presumably through a breach of the blood–ocular barrier and the release of normally sequestered antigens [3,4]. In most uveitis cases, however, the etiologic triggers are unknown and have been postulated to include antigenic mimicry by microorganisms in conjunction with a concomitant adjuvant effect, leading to the priming of effector T lymphocytes capable of recognizing ocular antigens [5]. Autoimmune uveitis is a sight-threatening inflammatory disorder that affects humans at all ages [1]. Current therapies for uveitis are largely based on immunosuppressive treatment, including corticosteroids, antimetabolites, and alkylating agents. Due to the nonspecific nature and the dose-limiting side effects of these drugs, the results of current treatment for autoimmune-mediated uveitis remain unsatisfactory [6]. Each year, 17.6% of active uveitis patients experience a transient or permanent loss of vision, and 12.5% of uveitis patients will develop glaucoma [7]. An improved understanding of uveitis pathogenesis is needed to develop effective treatments.

A robust model for human uveitis is experimental autoimmune uveitis (EAU) in mice, which can be induced by immunizing susceptible mouse strains with a retinal antigen, such as interphotoreceptor retinoid binding protein (IRBP) and retinal arrestin (retinal soluble antigen or S-antigen) [8]. IRBP functions to transport retinoids, which are essential for the visual cycle, between the retinal pigment epithelium and the photoreceptors. S-antigen is the visual arrestin that quenches photoactivated rhodopsin in the process of visual signal transduction. Both proteins are highly evolutionarily conserved and are major components of the photoreceptor cell layer. The retinal antigens that are involved in the visual cycle and that can serve as targets in EAU are typically unique not only to the eye but also to the whole body. The only other site of expression (within the limits of detection of currently available methods) is the pineal gland (“third eye”), which controls the circadian rhythm and shares many vision-related proteins with the retina [9].

During EAU progression, the infiltration of inflammatory cells into the retina and/or uvea begins approximately seven days after induction. The stages before and after day seven postinduction are defined as the early and amplification phases, respectively. In the early phase, the upregulation of inducible nitric oxide synthase (iNOS), which catalyzes the production of nitric oxide (NO), is detected in the photoreceptor mitochondria of retina [10,11,12]. Inflammation is a natural defense mechanism against pathogens and it is associated with many pathogenic diseases such as autoimmune diseases. Oxidative stress refers to the excessive production of reactive oxygen species (ROS) in the cells and tissues and antioxidant system may not be able to neutralize them, which can lead to chronic inflammation [13]. ROS are strong stimulators of the transcription factor nuclear factor kappa B (NF-κB), which increases the transcription of inflammatory cytokines and chemokines [14]. An increase in the oxidative stress response with the generation of ROS, superoxide and hydrogen peroxide was also found. The major source of these oxidants is nicotinamide adenine dinucleotide phosphate (NADPH) oxidase 2 (NOX2) [15,16]. NO and superoxide rapidly react to form the highly toxic peroxynitrite OONO^−^. NO and peroxynitrite are reactive nitrogen species (RNS). Oxidative stress induces the nitration of photoreceptor mitochondrial proteins and the peroxidation of membrane lipids. The ROS generated by oxidative stress and RNS are therefore proposed to be initial pathological events leading to the EAU-induced damage observed during the amplification phase. The role of ROS in EAU remains elusive.

The release of ROS and its downstream products from phagocytes, which is known as the respiratory burst, plays a significant role in fighting against invading pathogens. The importance of the innate immune defense with a functional phagocyte NOX2 is clearly exemplified in chronic granulomatous disease (CGD), a rare genetic disorder characterized by severe recurrent infections due to the inability of neutrophils and macrophages to mount a respiratory burst to kill invading pathogens [17]. In addition to recurrent and severe infections, inflammatory manifestations are also common in CGD patients, including the gastrointestinal tract (88.2%), lungs (26.4%), the urogenital tract (17.6%), and eyes (8.8%) [18,19]. NOX2 is composed of five subunits, including p47^phox^, which is also called neutrophil cytosolic factor 1 (Ncf1) [20]. Ncf1 is an essential component of the NOX2 complex because the absence of Ncf1 leads to undetectable NOX2 activity as measured by the ROS response of neutrophils in mice [21]. The second most common genetic defect, responsible for approximately 30% of CGD cases, is an autosomal recessive mutation in *Ncf1* [17]. Mice without Ncf1 display augmented disease severity in two models of autoimmune disorders, experimental autoimmune encephalomyelitis (EAE) provoked by native myelin oligodendrocyte glycoprotein (MOG) and arthritis caused by collagen or serum [21,22]. To address the role of ROS in EAU in vivo, we compared wild-type mice and Ncf1-deficient mice [23] and assessed treatment with N-acetylcysteine (NAC), an ROS inhibitor used in the clinic. Surprisingly, we discovered that the suppression of ROS due to Ncf1 deficiency or NAC treatment decreases EAU severity in mice.

## 2. Results

### 2.1. ROS Deficiency due to the absence of Ncf1 Decreases the Severity of EAU in Mice with Reduced Levels of Oxidative Stress, Inflammatory Mediators, and NF-ĸB Activation in the Retina

#### 2.1.1. Ncf1 Deficiency Reduces Malondialdehyde Levels in the Retinas and Spleens of Mice with EAU Induction

We monitored the effect of EAU on the stimulation of oxidative stress by measuring malondialdehyde, which serves as a marker for oxidative stress and is produced upon lipid peroxidation [24]. To induce EAU, we immunized wild-type mice with human IRBP peptide 1-20 (*n* = 14), which increased malondialdehyde levels in the retina and spleen after 21 days by more than 2-fold compared to those of naïve wild-type mice without immunization (*n* = 10) (Figure 1A,B). As Ncf1 is an essential component of NOX2 that generates ROS to promote oxidative stress [15,16], we therefore measured malondialdehyde in immunized *Ncf1^−/−^* mice (*n* = 7), in which a point mutation occurs at the 22 position of exon 8 to result in the aberrant splicing of transcripts and undetectable protein expression [25]. The absence of Ncf1 reduced the amount of malondialdehyde in the retina and spleen of immunized mice to levels comparable to those of wild-type mice without immunization (Figure 1). Collectively, IRBP immunization enhances oxidative stress, and ROS production contributes to oxidative stress in mouse tissues.

#### 2.1.2. Absence of Ncf1 Reduces EAU Severity in Mice

To further determine the role of Ncf1 in EAU, we monitored EAU in wild-type (*n* = 20), *Ncf1^+/−^* (*n* = 4), and *Ncf1^−/−^* (*n* = 20) mice for 28 days. IRBP progressively intensified the severity of EAU in wild-type mice with a peak disease score at 21 days after immunization (Figure 2A). Comparable disease scores were detected in immunized wild-type and *Ncf1^+/−^* mice. Notably, the EAU disease scores of *Ncf1^−/−^* mice were significantly lower than those of wild-type and *Ncf1^+/−^* mice after immunization. We performed hematoxylin-eosin staining on the mouse retina. Histologically, the retinas of naïve wild-type and *Ncf1^−/−^* mice without EAU induction were morphologically similar (Figure 2B). In wild-type mice with EAU induction for 21 days, the retina became thick with edema and leukocyte infiltration, and the retinal structure was disrupted with folds in the inner and outer nuclear layers. In *Ncf1^−/−^* mice with EAU induction, the retinal edema, leukocyte infiltration, and folds were absent. The retinal morphology of *Ncf1^−/−^* mice with EAU induction was comparable to that of naïve wild-type and *Ncf1^−/−^* mice without EAU induction. The retinal disease score and histology results are consistent and reveal that the absence of Ncf1 decreases EAU severity in mice.

#### 2.1.3. The Influence of Ncf1 Deficiency on the Levels of Inflammatory Mediators in the Retinas of Mice with EAU Induction

Both EAU and ROS can amplify the expression of cytokines and chemokines to affect EAU severity. We investigated the influence of EAU and Ncf1 deficiency on the expression of cytokines and chemokines in the mouse retina. Cytokines (interleukin (IL)-1β, interferon (IFN)-γ, and tumor necrosis factor (TNF)-α) can promote the ROS response [26]. During EAU, leukocytes, macrophages and T lymphocytes can infiltrate the retina [10]. Monocyte chemoattractant protein (MCP)-1 is a chemokine capable of recruiting both macrophages and lymphocytes [27]. Th1, Th2, and Th17 responses regulate EAU [28]. We therefore assessed MCP-1, Th1 cytokines (IFN-γ and IL-12), Th2 cytokines (IL-4 and IL-6), and Th17 cytokines. EAU induction increased the levels of cytokines (IL-1α, IL-1β, IL-4, IL-6, IL-12, IL-17, and TNF-α) and the chemokine (MCP-1), but not IFN-γ, in the retina of wild-type mice after 21 days (Figure 3A–I). The absence of Ncf1 failed to increase IL-1α, IL-1β, IL-4, IL-6, IL-12, IL-17, TNF-α, and MCP-1 levels in the retina of mice with EAU induction. We also performed immunohistochemical staining on the mouse retina to detect IL-1β, TNF-α, and MCP-1, which were below the detection limit in naïve wild-type mice without EAU induction (Figure 3J). Abundant IL-1β, TNF-α, and MCP-1 were detected mostly in the inner nuclear layer of retina of wild-type mice 21 days after EAU induction. The absence of Ncf1 diminished the expression of IL-1β, TNF-α, and MCP-1 in the retina of mice with EAU induction. In summary, the absence of Ncf1 suppresses the expression of inflammatory mediators in the retina of mice with EAU induction.

#### 2.1.4. Absence of Ncf1 Reduces NF-κB Activation in the Retinas of Mice with EAU Induction

Both EAU and ROS can augment the expression and activation of the transcription factor NF-κB [6,29], which promotes the expression of cytokines and chemokines, such as IL-1β, IL-6, IL-17, TNF-α, and MCP-1 [30,31]. After NF-κB is activated, its components, p65 and p50, are translocated to the nucleus [32]. We performed immunofluorescence staining to detect p65 and nuclei. The expression of p65 was below the detection limit in the retina of naïve wild-type mice without EAU induction (Figure 4A). In wild-type mice with EAU induction for 21 days, abundant p65 was detected in the retina, and p65 was located in the nuclei of some retinal cells (Figure 4A). The absence of Ncf1 reduced p65 expression and nuclear translocation in the retina of mice with EAU induction. We also extracted nuclear proteins from mouse retinas and assessed NF-κB activity by measuring the binding of NF-κB to its response element using electrophoretic mobility shift assay (EMSA). The levels of NF-κB activity were low in both naïve wild-type and *Ncf1^−/−^* mice without EAU induction (Figure 4B). EAU enhanced the level of NF-κB activity in wild-type mice 21 days after induction. However, the level of NF-κB activity in *Ncf1^−/−^* mice with EAU induction was low and comparable to that of naïve wild-type and *Ncf1^−/−^* mice without EAU induction. The NF-κB immunofluorescence and EMSA results obtained from mouse retinas suggest that EAU increases NF-κB activation and that the absence of Ncf1 inhibits NF-κB activation.

### 2.2. Treatment with the Antioxidant NAC Reduces the Severity of EAU in Mice with Reduced Levels of Oxidative Stress, Inflammatory Mediators, and NF-ĸB Activation in the Retina

#### 2.2.1. NAC Treatment Reduces EAU Severity in Mice

As our results suggest that ROS exacerbate EAU progression, we next investigated whether treatment with an ROS inhibitor can improve EAU. NAC is the precursor of the ROS scavenger glutathione [33] and has been tested for the treatment of chronic bronchitis in patients [34]. NAC is also used for antioxidation and detoxification in the clinic [35]. NAC at a dose of 150 mg/kg can reduce the disease phenotypes not related to the eye in rats and mice [36,37,38], so we assessed the effect of NAC treatment on EAU in wild-type mice using this dose. Compared with phosphate-buffered saline (PBS) treatment (*n* = 21), NAC treatment (*n* = 21) significantly decreased mouse EAU disease scores (Figure 5A). The body weights of mice with EAU induction and treated with NAC or PBS were not significantly different (Figure 5B), suggesting that the side effect of NAC is minimal. Histological staining results showed that the retinal folds, damage, and leukocyte infiltration in mice with EAU induction were decreased with NAC treatment compared with PBS treatment (Figure 5C).

#### 2.2.2. The Influence of NAC Treatment on the Levels of Inflammatory Mediators and NF-κB Activation in the Retinas of Mice with EAU Induction

Mouse tissues were harvested 21 days after EAU induction for analyses. NAC treatment significantly decreased malondialdehyde levels in the retina and spleen of mice with EAU induction (Figure 1C,D), indicating that NAC suppresses the oxidative stress response of EAU mice. NAC treatment diminished the levels of IL-1α, IL-1β, IL-4, IL-6, IL-12, IL-17, TNF-α, and MCP-1, but not that of IFN-γ, in the retina of mice with EAU induction (Figure 6A–I). Immunohistochemical staining results revealed that NAC treatment inhibited the expression of IL-1β, TNF-α, and MCP-1 in the retina of mice with EAU induction (Figure 6J). The immunofluorescence staining results showed that NAC treatment diminished p65 expression and nuclear translocation in the retina of mice with EAU induction (Figure 7). EMSA results showed that NAC treatment reduced the level of NF-κB activity in the retina of mice with EAU induction (data not shown). The NF-κB immunofluorescence and EMSA results obtained from mouse retinas reveal that NAC treatment prevents NF-κB activation.

## 3. Discussion

The results of Ncf1 deficiency and NAC treatment are consistent and collectively show that the suppression of ROS decreases EAU severity in mice. The identification of a single-nucleotide mutation in the mouse *Ncf1* gene that leads to a reduced oxidative burst and suppressed EAU response is indeed a surprising and challenging finding. Our observation that Ncf1-deficient mice are protected from IRBP peptide-induced EAU is difficult to reconcile with the findings of aggravated diseases detected in the same mice induced to develop two other models of autoimmune disorders, EAE induced by MOG and arthritis caused by collagen or serum [21]. Therefore, the *Ncf1* gene has a more general effect on T cell-dependent autoimmune diseases. The finding that Ncf1-deficient mice showed exacerbated EAE [21] contrasts several reports demonstrating that NAC treatment inhibits the disease in mice and rats [36,39,40]. Interestingly, in Ncf1-deficient mice with EAE induction, the native MOG protein enhanced disease severity, but the immunodominant peptides reduced disease severity [21,22]. This finding indicates that the immune recognition of native epitopes or the uptake and processing of protein antigens is important for Ncf1 function because the administration of the peptide will bypass these steps before binding to MHC class II molecules on antigen-presenting cells [22]. This hypothesis may partially explain why Ncf1-deficient mice are resistant to MOG peptide-induced EAE and, as in our study, IRBP peptide-induced EAU.

During EAU, TNF-α, mitochondrial DNA damage, and macrophages can induce the ROS response [10,11,12,41]. The abundant infiltration of leukocytes, such as macrophages followed by T lymphocytes, into the retina is detected more than 5 days after induction [10,11]. Retinal cells are likely to be the initial sources of TNF-α and mitochondrial DNA damage, which are increased 3 and 4 days after induction, respectively [12,41]. In the amplification phase of EAU, when the disease score reached a peak in the mouse retina 21 days after induction, we found elevated levels of ROS, which are likely produced by both retina cells and infiltrating leukocytes. Indeed, ROS inducers (TNF-α and IL-1β) and a macrophage chemokine (MCP-1) were detected in the damaged mouse retina with both retina cells and infiltrating leukocytes after EAU induction. RNS can also elicit ROS and has been proposed to initiate EAU with ROS [10,11,12]. However, the RNS producer iNOS is reported to be dispensable for EAU pathogenesis, as iNOS knockout mice develop EAU with scores and cellular responses similar to wild-type mice after IRBP immunization [42]. Nevertheless, a highly selective inhibitor of iNOS enhances the EAU induced by S-antigen [43]. We also found that unlike other cytokines, IFN-γ was not induced in the EAU group. Rajendram, R. et al. found that IFN-γ transcripts were significantly upregulated from day 5 to day 14 in the retinas of EAU mice [10]. However, we assess IFN-γ on day 21 after EAU induction. Therefore, the possibility that the level of IFN-γ in the retinas of EAU mice has returned to normal after three weeks of EAU induction cannot be ruled out. Besides, IFN-γ-deficient mice on the C57BL/6 background are equally susceptible to IRBP-induced EAU as wild-type mice [44]. Th17 cells, and their related cytokines, such as IL-6 and IL-17, are likely to be more important inflammatory mediators in autoimmune uveitis [45].

A report demonstrated that NAC reduces the disease score and histopathology of IRBP-induced EAU in mice without details on the mechanism [46]. The same report also tested two kinds of structurally related mineral oils with pro-oxidative effects to treat EAU in mice. Paradoxically, phytol failed to affect EAU, but pristane, which can induce autoimmune diseases (rheumatoid arthritis and lupus) [47,48], protected mice from EAU. Since we found that ROS production and EAU severity were decreased in Ncf1-deficient mice compared with wild-type mice, we assessed the effect of treatment with NAC, which could decrease ROS production. We found that NAC treatment in IRBP-immunized wild-type mice significantly ameliorated the clinical course of the EAU response. The capacity of NAC to ameliorate inflammatory disease was also reported in rats and mice with EAE [40]. In the rat study, NAC treatment reduced TNF-α, IL-1β, IFN-γ, and iNOS levels in the central nervous system and attenuated EAE induced by myelin basic protein [39]. The EAE studies indicate that NAC treatment may be of therapeutic value in multiple sclerosis against the inflammatory process associated with the infiltration of activated mononuclear cells into the central nervous system [40]. Our additional findings that NAC has the capacity to suppress oxidative stress, the expression of chemokines and proinflammatory cytokines, the transmigration of inflammatory cells into the retina of EAU mice, and NF-ĸB activation identify a potential drug for the treatment of human uveitis in the future.

## 4. Materials and Methods

### 4.1. EAU Induction and NAC Treatment in Mice

C57BL/6J mice and C57BL/6J-derived mice deficient in Ncf1 (B6(Cg)-*Ncf1^m1J^*/J, No. 004742) were purchased from the Jackson Laboratory (Bar Harbor, ME, USA) and bred in our college animal center. All mouse experiments were performed in compliance with a protocol approved by the Institutional Animal Care and Use Committee of National Cheng Kung University (with the approval number of 101030) and with the statement of the Association for Research in Vision and Ophthalmology (ARVO) for the Use of Animals in Ophthalmic and Vision Research. To induce EAU, 8- to 12-week-old female mice were immunized with human IRBP peptide 1-20 (GPTHLFQPSLVLDMAKVLLD) in complete Freund’s adjuvant containing inactivated *Mycobacterium tuberculosis* H37RA (Difco Laboratories, Detroit, MI, USA) at two sites on the lower back, followed by an intraperitoneal injection of pertussis toxin as previously described [6]. Wild-type mice were treated with one administration of NAC (Sigma-Aldrich, Saint Louis, MO, USA) at a dose of 150 mg/kg or PBS every other day by intraperitoneal injection starting one hour before EAU induction. EAU scoring was performed by examining the ocular fundus of mouse eyes with a slit lamp. The severity of inflammation was graded on the following scale of 1–5 as previously described [6]: 0, no inflammation; 1, ≤5 focal vasculitis spots or soft exudates; 2, linear vasculitis or spotted exudates in <50% of the retina; 3, linear vasculitis or spotted exudates in ≥50% of the retina; 4, retinal hemorrhage or severe exudates and vasculitis; and 5, exudative retinal detachment or subretinal (or vitreous) hemorrhage. The severity of uveitis is represented as the highest clinical score achieved by either eye in a mouse. After EAU induction for 21 days, mouse eyes and spleens were harvested, and retinas were extracted for assays.

### 4.2. Malondialdehyde Assays

The eyes were enucleated from euthanized mice. The eyeballs were cut at the equator around the ora serrata, and the posterior pole of the eyes was separated from the anterior pole and lens. From the posterior pole, the neurosensory retina was extracted from retinal pigment epithelial layer. The extract from six retinas was placed in 300 μL of 0.5% NP-40 (Abcam, Cambridge, UK) on ice (one minute) and briefly sonicated five times for 10 s (MicrosonTM XL2000 Ultrasonic liquid processor, Qsonica, LLC, Newton, CT, USA). After removal of the insoluble material by centrifugation (200× *g* for 5 min), the protein concentration of the retinal extract was measured at 280 nm on ND-1000 Spectrophotometer as previously described [6]. Spleens were processed in the same manner as retinas. Briefly, one spleen was placed in 300 μL of 0.5% NP-40, ground, sonicated, and centrifuged. The protein concentration of supernatants was measured at 280 nm by a ND-1000 spectrophotometer. Retina and spleen lysate supernatants were subjected to malondialdehyde and Luminex assays. Quantification of malondialdehyde was performed with a kit (NWLSS^TM^ Malondialdehyde Assay, Northwest Life Science Specialties, LLC, Vancouver, WA, USA).

### 4.3. Luminex Assay

Quantification of IL-1α, IL-1β, IL-4, IL-6, IL-12, IL-17, IFN-γ, TNF-α, and MCP-1 in retinal lysate supernatants was carried out using murine multiplexing bead immunoassays (Invitrogen, Carlsbad, CA, USA) according to manufacturer’s instruction. Briefly, 25 μL of retinal samples in PBS was incubated with antibody-coupled beads. After a series of washes, a biotinylated detection antibody was added to the beads, and the reaction mixture was detected by the addition of streptavidin-phycoerythrin. The bead set was analyzed using a flow-based Luminex 200 suspension array system (Invitrogen, Carlsbad, CA, USA).

### 4.4. Histological, Immunohistochemical, and Immunofluorescence Staining

The eyes were processed, sectioned, and stained with hematoxylin and eosin as previously described [6]. Sections were also subjected to immunohistochemical staining with antibodies against mouse IL-1β, TNF-α, or MCP-1 (Abcam, Cambridge, MA, USA). Additionally, sections were subjected to immunofluorescence staining with DAPI (4′,6-diamidino-2-phenylindole) for DNA and the antibody against mouse p65 (Abcam, Cambridge, MA, USA).

### 4.5. EMSA of NF-κB

Retinas were processed to extract nuclear proteins, and the protein concentration of the samples was determined by a bicinchoninic acid assay kit (Pierce Biotechnology, Rockford, IL, USA) as previously described [6]. EMSA was performed with a kit (NF-κB DNA-binding protein detection system, Pierce Biotechnology, Rockford, IL, USA) according to the manufacturer’s instructions as previously described [6]. Briefly, nuclear protein (10 μg) was incubated with a biotin-labeled NF-κB consensus oligonucleotide probe (5′-AGTTGAGGGGACTTTCCCAGGC-3′). The specificity of the DNA protein binding was determined by adding a 100-fold molar excess of unlabeled NF-κB oligonucleotide for competitive binding 10 min before adding the biotin-labeled probe.

### 4.6. Statistical Analyses

Data are expressed as the mean ± SEM. For statistical comparison, EAU disease scores were analyzed by the Wilcoxon signed-rank test, and the remaining data were analyzed by the Mann–Whitney U test using Prism 8.0 software. A *p* value of <0.05 was considered statistically significant.

## 5. Conclusions

In conclusion, ROS play an important role in the pathogenesis of EAU in mice. NAC treatment with ameliorated ROS production seems to be a novel therapy for autoimmune uveitis in human in the future.

## Figures and Tables

**Figure 1 ijms-21-03261-f001:**
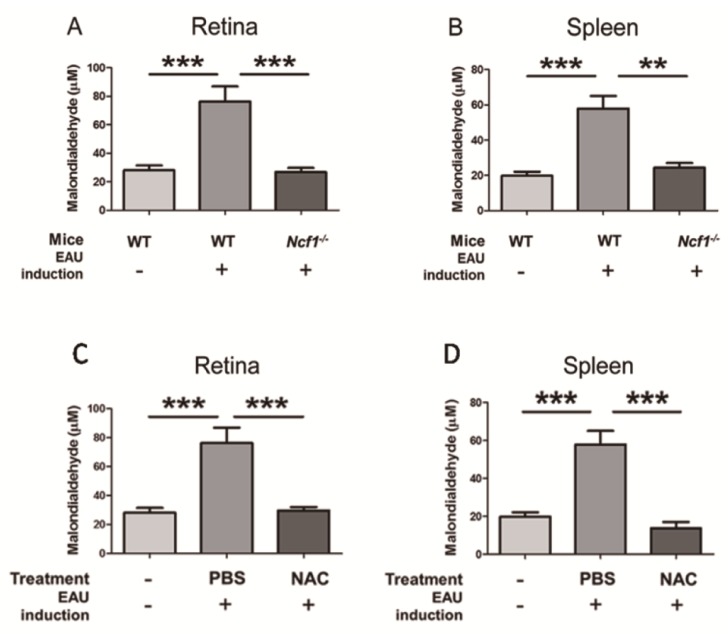
Ncf1 deficiency and N-acetylcysteine (NAC) treatment reduce malondialdehyde levels in the retinas and spleens of mice with experimental autoimmune uveitis (EAU) induction. (**A**,**B**) Malondialdehyde levels in retinas and spleens of wild-type (WT) mice without (-) EAU induction (*n* = 10) or WT mice (*n* = 14) and *Ncf1^−/−^* mice (*n* = 7) with (+) EAU induction are shown. (**C**,**D**) Malondialdehyde levels in retinas and spleens of wild-type mice without (-) EAU induction (*n* = 10) and wild-type mice with (+) EAU induction and treated with phosphate buffered saline (PBS) (*n* = 14) or NAC (*n* = 10) are shown. Data show the mean + SE values (error bars). ** *p <* 0.005, *** *p <* 0.001.

**Figure 2 ijms-21-03261-f002:**
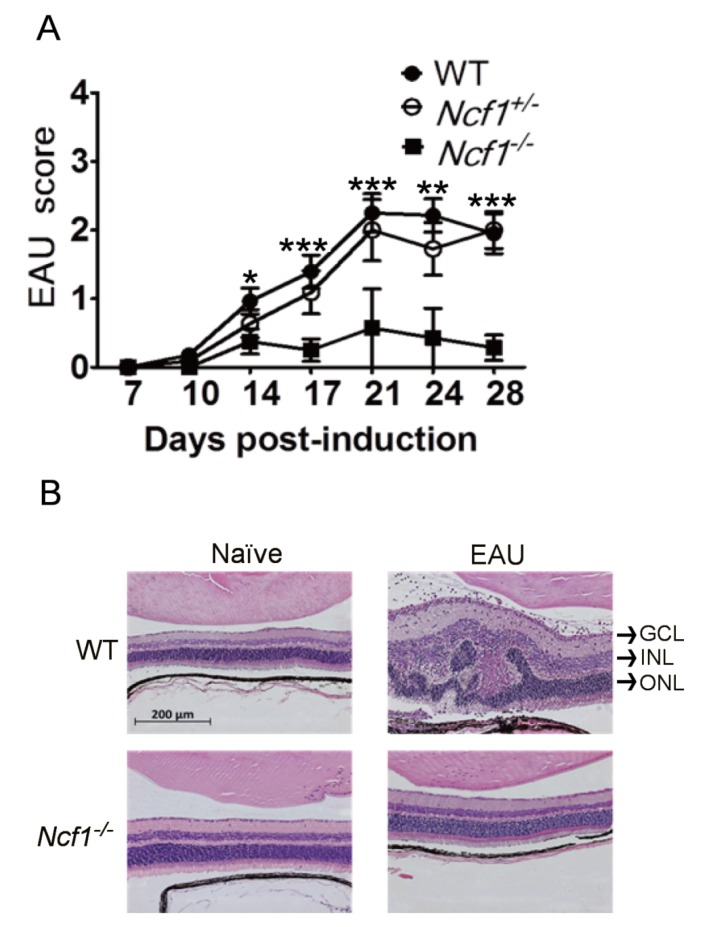
Absence of Ncf1 reduces EAU severity in mice. (**A**) Disease scores of wild-type mice (WT, *n* = 20), *Ncf1^+/−^* mice (*n* = 4), and *Ncf1^−/−^* mice (*n* = 20) after EAU induction are shown. Data show the mean ± SE values (error bars). * *p* < 0.05, ** *p* < 0.005, *** *p <* 0.001 compared with the *Ncf1^−/−^* mice at the same time point. (**B**) Eyes of WT and *Ncf1^−/−^* mice without (naïve) or with EAU induction were subjected to H&E staining. The retina portion is shown. Images are representative of at least four samples per group from two independent experiments. GCL, ganglion cell layer; INL, inner nuclear layer; ONL, outer nuclear layer.

**Figure 3 ijms-21-03261-f003:**
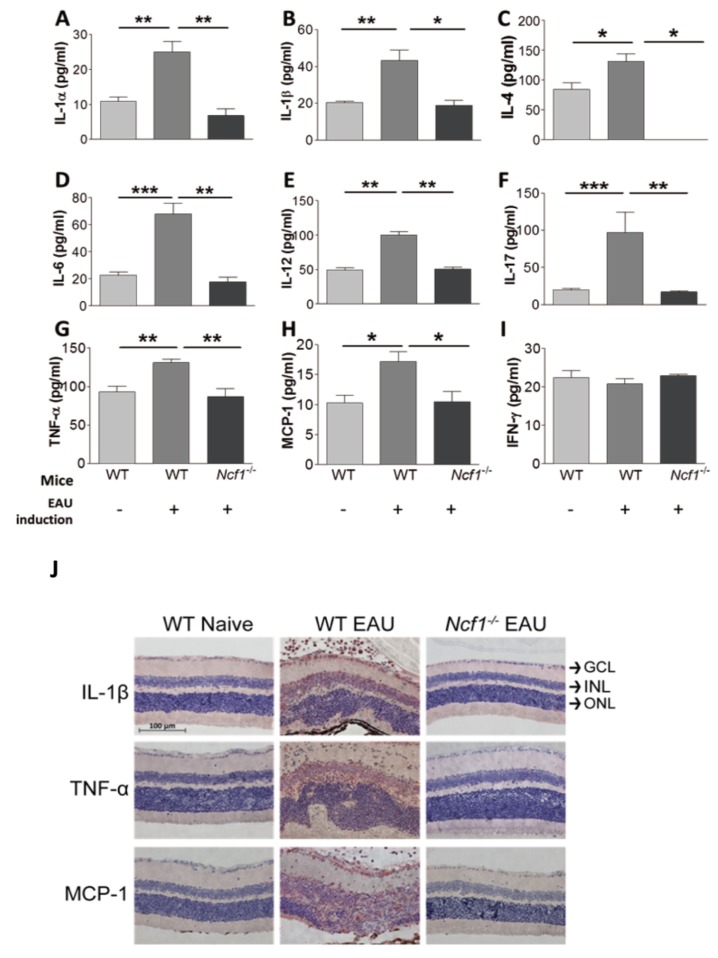
The influence of Ncf1 deficiency on the levels of inflammatory mediators in the retinas of mice with EAU induction. Levels of interleukin (IL)-1α (**A**), IL-1β (**B**), IL-4 (**C**), IL-6 (**D**), IL-12 (**E**), IL-17 (**F**), tumor necrosis factor (TNF)-α (**G**), monocyte chemoattractant protein (MCP)-1 (**H**), and interferon (IFN)-γ (**I**) in the retinas of wild-type (WT) and *Ncf1^−/−^* mice with (+) or without (-) EAU induction are shown. Data show the mean + SE values (error bars) of at least five samples per group. * *p* < 0.05, ** *p* < 0.005, *** *p* < 0.001. (**J**) Eyes of WT and *Ncf1^−/−^* mice without (naïve) or with EAU induction were subjected to staining for the indicated inflammatory mediators. The retina portion is shown. Images are representative of at least four samples per group from two independent experiments. GCL, ganglion cell layer; INL, inner nuclear layer; ONL, outer nuclear layer.

**Figure 4 ijms-21-03261-f004:**
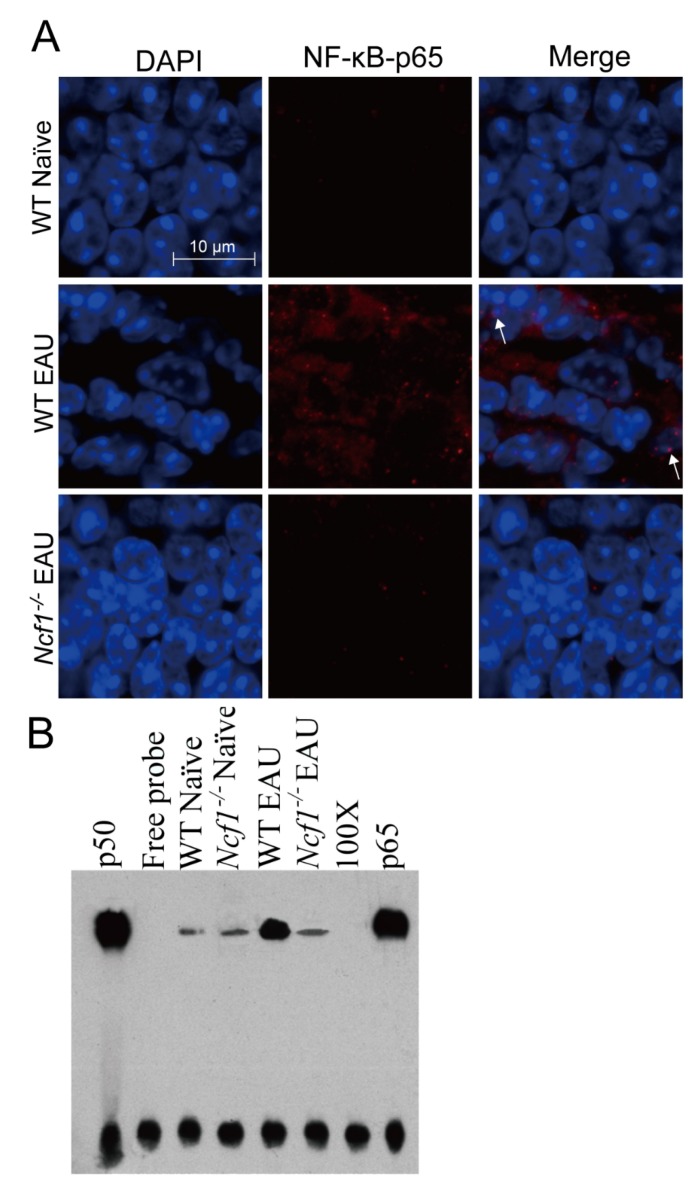
Absence of Ncf1 reduces NF-κB activation in the retinas of mice with EAU induction. Eyes and retinas of wild-type (WT) and *Ncf1^−/−^* mice without (naïve) or with EAU induction were subjected to immunofluorescence staining with DAPI for nuclei and with antibody for NF-κB (p65) (**A**) and to electrophoretic mobility shift assay (EMSA) (**B**). In panel A, the outer nuclear layer of the retina is shown. Images are representative of at least four samples per group from two independent experiments. Arrows indicate NF-κB in the nucleus. In panel B, lane 1: p50 subunit of NF-κB; lane 2: free probe; lane 3: wild-type naïve mice; lane 4: *Ncf1^−/−^* naïve mice; lane 5: wild-type EAU mice; lane 6: *Ncf1^−/−^* EAU mice; lane 7: 100-fold molar excess of unlabeled NF-κB probe. Lane 8: biotinylated probe with anti-p65 antibody. Top bands are the complexes of p50/p65/biotin-labeled DNA probe and bottom bands are free probe.

**Figure 5 ijms-21-03261-f005:**
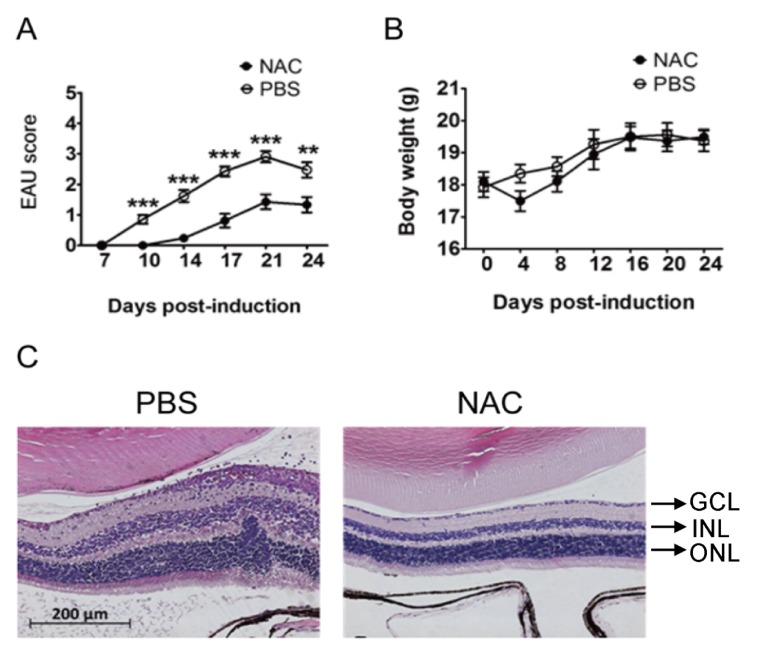
NAC treatment reduces EAU severity in mice. Disease scores (**A**) and body weights (**B**) of wild-type mice with EAU induction and treated with PBS (*n* = 21) or NAC (*n* = 21) are shown. Data show the mean ± SE values (error bars). ** *p* < 0.005, *** *p* < 0.001 compared with the NAC treatment group at the same time point. (**C**) Eyes of mice described above were subjected to staining. The retinal portion is shown. Images are representative of at least four samples per group from two independent experiments.

**Figure 6 ijms-21-03261-f006:**
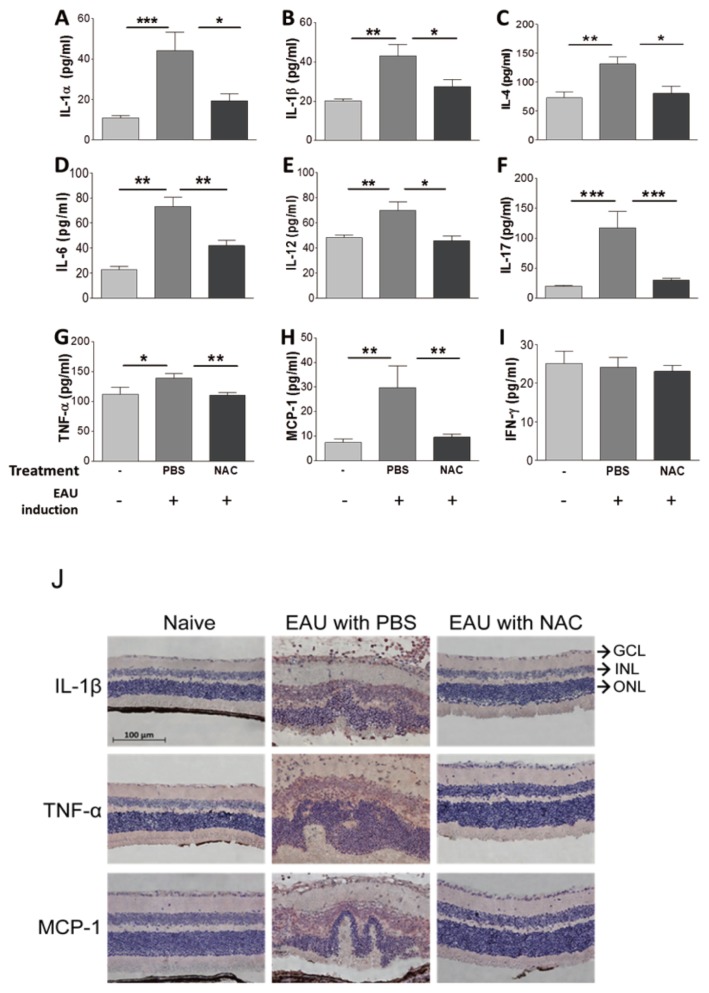
The influence of NAC treatment on the levels of inflammatory mediators in the retinas of mice with EAU induction. Levels of IL-1α (**A**), IL-1β (**B**), IL-4 (**C**), IL-6 (**D**), IL-12 (**E**), IL-17 (**F**), TNF-α (**G**), MCP-1 (**H**), and IFN-γ (**I**) in retinas of wild-type with (+) or without (-) EAU induction and treated with PBS or NAC are shown. Data show the mean + SE values (error bars) of > eight samples per group. * *p* < 0.05, ** *p* < 0.005, *** *p* < 0.001. (**J**) Eyes of wild-type mice without EAU induction (naïve) or with EAU induction and treated with PBS or NAC were subjected to staining for the indicated inflammatory mediators. The retinal portion is shown. Images are representative of at least four samples per group from two independent experiments. GCL, ganglion cell layer; INL, inner nuclear layer; ONL, outer nuclear layer.

**Figure 7 ijms-21-03261-f007:**
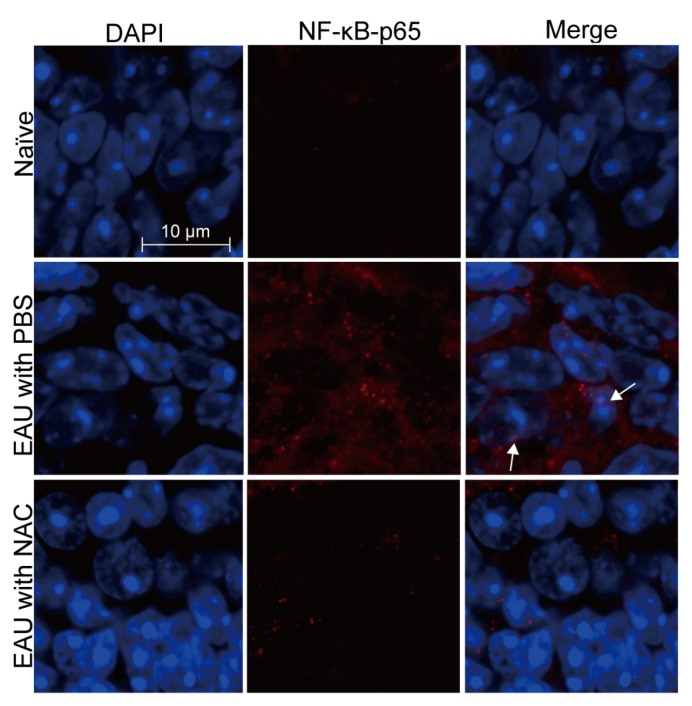
NAC treatment reduces NF-κB activation in the retinas of mice with EAU induction. Eyes and retinas of wild-type mice without EAU induction (naïve) or with EAU induction and treated with PBS or NAC were subjected to immunofluorescence staining for nuclei with DAPI and for NF-κB (p65) with antibody. The outer nuclear layer of the retina is shown. Images are representative of at least four samples per group from two independent experiments. Arrows indicate NF-κB in the nucleus.

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
