# Peer review of "Suppression of the Reactive Oxygen Response Alleviates Experimental Autoimmune Uveitis in Mice"

_ijms, 2020, doi:10.3390/ijms21093261_

Round 1

Reviewer 1 Report

The Authors have submitted an interesting and well presented study, some minor revisions may improve the understanding of their results.

For clarity of the paper the “Materials and Methods” section should be moved after the “Introduction” section and before the “Results” section; moreover, in the “Materials and Methods” section some lines about the number of mice evaluated and treated should be written, not only as a legend to Figures 1 and 2 (example: line 215: “treated with PBS (n = 14) or NAC (n = 10)”)

In lines 42-43 it is said that “uveitis occurs in approximately 0.54% of the population with idiopathic uveitis”. This phrase does not make much sense in the present form. I think that “uveitis occurs in approximately 0.54% of the population” and that “approximately 30% of cases of uveitis are idiopathic” are the phrases the Authors wanted to link together.

“During EAU” is repeated twice in lines 285-286.

Line 290-291: “we found an elevated ROS levels”; structure of the phrase might be improved.

Line 48: “normally sequesters antigens” should be changed to “normally sequestered antigens”

Reviewer 2 Report

Hsu et al. investigated the role of reactive oxygen species (ROS) in experimental autoimmune uveitis (EAU) using ROS-deficient (Ncf1-/-) mice. Overall, this topic is well chosen and surely of interest in the field, as well as worth to investigate. However, this article surely needs be improved in contents as well as structure, and I have several concerns, as follow. I hope these comments help improve the quality of this article.

  1. EAU is an inflammatory disease, and ROS is one of the main inflammatory mediators in the inflammatory responses. Therefore, the authors need to describe inflammation (inflammatory responses) and the relevance of ROS with inflammation in the Introduction section.
  2. Fig. 2A: Please compare the statistical significance (difference) between WT and experimental groups at every day point (day 7, 10, 14, 17, 21, 24, and 28).
  3. Please divide 4.2. section into two sections – Malondialdehyde assay and Luminex assay. It is unclear how the authors conducted Luminex assay (such as how the authors prepared tissue lysates and used tissue lysates for this assay). Please describe the assay procedure in more detail.
  4. Please provide the possible reasons why unlike other cytokines, IFN-gamma was not induced in EAU group (Fig. 3I and 6I) in the discussion section.
  5. Fig. 4B is unclear. What are the bands of all groups? Are they p50 or p65? NF-kB consists of p50 and p65, therefore, the authors need to make sure what these bands are (p50 or p65).
  6. Fig. 4B: Generally, NF-kB activation is evaluated by its phosphorylation and nuclear translocation, not just its expression. The authors examined only NF-kB-p65 expression in Fig. 4A, but need to provide more evidence of NF-kB activation, such as NF-kB-p65 phosphorylation and nuclear translocation to support the authors’ hypothesis. It seems NF-kB-p65 is nuclear translocated in the Fig. 4A ‘Merge’ panel, but NF-kB-p65 phosphorylation needs to be provided. Moreover, NF-kB is unclear in Fig. 4A, and NF-kB needs to be corrected to NF-kB-p65. This also needs to be applied to Fig. 7.
  7. Please provide the animal study approval number or protocol number for this study.
  8. Please go over the entire manuscript carefully and correct all the typos and grammatical errors.

Round 2

Reviewer 2 Report

The authors appropriately addressed the reviewer's concerns, and this article is now acceptable for publication.